# The Impact of Climate Change on the Mental Health of Populations at Disproportionate Risk of Health Impacts and Inequities: A Rapid Scoping Review of Reviews

**DOI:** 10.3390/ijerph21111415

**Published:** 2024-10-25

**Authors:** Germán Andrés Alarcón Garavito, Lina Fernanda Toncón Chaparro, Sarah Jasim, Francesca Zanatta, Ioanna Miliou, Maria Bampa, Gesche Huebner, Tara Keck

**Affiliations:** 1Rapid Research Evaluation and Appraisal Lab (RREAL), University College London, London W1W 7TY, UK; german.alarcon.20@ucl.ac.uk; 2Department of Targeted Interventions, University College London, London W1W 7TY, UK; lina.chaparro.21@ucl.ac.uk; 3Care Policy and Evaluation Centre, London School of Economics and Political Science, London WC2A 2AE, UK; s.jasim@lse.ac.uk; 4Department of Primary Care and Population Health, University College London, London WC1E 7HB, UK; f.zanatta@ucl.ac.uk; 5Department of Computer and Systems Sciences, Stockholm University, SE-10691 Stockholm, Sweden; ioanna.miliou@dsv.su.se (I.M.); maria.bampa@dsv.su.se (M.B.); 6European Centre for Environment and Human Health, University of Exeter, Cornwall TR10 9FE, UK; g.huebner@exeter.ac.uk; 7Department of Neuroscience, Physiology and Pharmacology, University College London, London WC1E 6DE, UK

**Keywords:** climate change, mental health, populations at disproportionate risk of health impacts and inequities, literature review

## Abstract

The impacts of climate change on mental health are starting to be recognized and may be exacerbated for populations at disproportionate risk of health impacts or inequalities, including some people living in low- and middle-income countries, children, indigenous populations, and people living in rural communities, among others. Here, we conduct a rapid scoping review of reviews to summarize the research to date on climate impacts on the mental health of populations at disproportionate risk. This review highlights the direct and indirect effects of climate change, the common mental health issues that have been studied related to climate events, and the populations that have been studied to date. This review outlines key gaps in the field and important research areas going forward. These include a need for more systematic methodologies, with before-and-after comparisons or exposure/non-exposure group comparisons and consistent mental health outcome measurements that are appropriately adapted for the populations being studied. Further research is also necessary in regard to the indirect effects of climate change and the climate effects on indigenous populations and populations with other protected and intersecting characteristics. This review highlights the key research areas to date and maps the critical future research necessary to develop future interventions.

## 1. Introduction

Climate change is considered to be one of the biggest challenges of current times and affects many aspects of life [1], including environmental conditions, livelihoods, and health. Climate change also has significant implications for mental health and well-being, including with respect to how people interact with their communities and societies [2,3,4]. Understanding the multifaceted impacts of climate change on mental health involves considering both its direct and indirect effects [3,5]. The direct impacts of climate change on mental health are often studied in relation to the aftermath of extreme weather events, such as floods, wildfires, or storms, which can have an immediate impact on mental health [6]. Specifically, individuals who have lost their homes or loved ones due to extreme weather events may suffer from post-traumatic stress disorder (PTSD) or other forms of psychological distress [6]. Climate change also affects mental health indirectly [5,7], extending beyond immediate effects. But because many longer-lasting climate events, such as droughts or changes in wildlife patterns, are not discrete, it can be more difficult to link mental health effects to these events. The climate change impacts from these types of events can include the displacement of communities and the loss of social support networks, which may further exacerbate mental health issues [5,6,7,8]. These indirect effects may also be rooted in changes in socioeconomic structures or environmental degradation. For example, climate-change-induced droughts or crop failures leading to the loss of livelihood and economic stability can increase anxiety, depression, and feelings of hopelessness among affected individuals [9].

It is important to note that not all people’s mental health is affected by climate change equally. Certain populations at disproportionate risk of adverse health impacts and inequities, due to a combination of social, economic, and environmental factors, may also face an increased risk of mental health impacts from climate change [4,10], but research into these groups is sparse. These populations could include, but are not limited to, racial or ethnic minorities, children, older adults, individuals experiencing socioeconomic disadvantages, or individuals contending with specific medical and mental health conditions [3]. While each factor alone may not signify a disproportionate risk of health impacts or inequities per se, these factors’ cumulative impact can create risk for individuals and communities, which may be compounded by climate change stressors [3].

There are often disparities in the support for mental health among populations at disproportionate risk, and these groups may be left out of discussions about the impacts of climate change. Supporting these populations’ mental health requires not only the provision of accessible and culturally sensitive mental health services but also comprehensive social, economic, and environmental policies that foster resilience within populations and for individuals. Furthermore, any interventions for a given population need to be co-produced with and targeted for the specific needs of said population [5]. The urgency of this issue has sparked a growing body of literature and research endeavors aimed at understanding the climate change–mental health nexus with the goal of developing interventions to provide mental health support. As a result of this urgency, we conducted a rapid scoping review of reviews aiming to summarize existing research areas that have been focused on to date and identify the research gaps that need to be addressed to develop future strategies for mitigating these climate-related impacts on the mental health of populations at disproportionate risk.

## 2. Methods

The initial concepts and ideas for this review were discussed at an international mental health and climate change workshop in Stockholm, where the scope and methods were discussed and agreed upon by the co-authors. The review design was informed by guidance for conducting reviews of reviews [11]. We also followed the Preferred Reporting Items for Systematic Reviews and Meta-Analysis (PRISMA) statement and used it to guide the review design and report the methods and findings [12].

### 2.1. Search Strategy

We identified search terms by exploring keywords and terms from previous related publications [1,4,13] and incorporating co-authors’ experience-based suggestions. We limited our search strategy to a smaller number of terms given the rapid nature of this scoping review of reviews, a process that may have imposed some limitations on our results (see Section 4). We adopted a definition of climate change that encompassed the terms “climate change”, “global warming”, “extreme weather”, “extreme events”, “heat wave(s)”, and “sea level rise”. Mental health was broadly defined to include not only mental health disorders but also emotional states, leading to search terms like “eco-anxiety”, “eco-agency”, “eco-advocacy”, “solastalgia”, “climate grief”, “mental health”, “depression”, “anxiety”, “suicide”, “PTSD”, “worry”, “hope”, “fear”, “anxiety”, “anger”, “despair”, “sadness”, “hopelessness”, and “guilt”. Search terms for groups at disproportionate risk included “vulnerable population”, “underserved population”, “disadvantaged population”, “minority groups”, and “indigenous people”. The search terms for populations at disproportionate risk were broad to avoid defining who comprises these populations and instead identify groups the research conducted to date has identified. We tested and interchanged the terms by conducting exploratory searches on PubMed (Appendix A). The search strategy focused on four categories (mental health, climate change, populations at disproportionate risk, and review). Final searches were conducted in June 2023 on three databases: PubMed, PsycINFO, and EMBASE.

The search was limited to articles published between January 2013 and June 2023 and written in the English language. There were no location limitations because previous reviews had maintained a global scope in their searches. The selected timeframe is likely to cover a majority of the relevant work on the topic, given that the mental health impacts of climate change have only started to receive attention as an important research topic in the last few years. For instance, both the Lancet Countdown mental health indicator [14] and the WHO’s first policy brief on this topic [15] were published in 2021 and 2022, respectively.

### 2.2. Document Selection

The search results were imported into Rayyan, a validated tool with semi-automated features enabling the detection of duplicated publications from the different databases [16]. Title- and abstract-level screening was conducted independently by one researcher (LF-TC). Following the initial screening at the title/abstract level, one researcher (GA-AG) cross-checked 10% of exclusions against the inclusion criteria. The remaining publications that met the inclusion criteria were organized to continue full-text screening for eligibility. In this phase, 100% of the included papers and 25% of the excluded papers were reviewed by two researchers (GA-AG and LF-TC), and discrepancies were resolved via discussion until consensus was reached. While a variety of definitions of mental health and well-being are present in the literature, we followed the structured framework for defining well-being [8]. This understanding considers the dynamic interplay between individuals, families, communities, and societies and the multiple circumstances that they experience. We followed the previously established healthcare approach to populations at disproportionate risk of health impacts and inequities, which defines these populations as including factors such as ethnicity, race, gender, age, culture, language, occupation, religion, education, socioeconomic status, medical condition status, or living with disability [17]. Detailed inclusion and exclusion criteria are provided in Table 1.

### 2.3. Data Extraction

Data extraction was conducted by using an extraction Google Sheets database to organize the review process. The extraction form was first piloted by two researchers (GA-AG and LF-TC), and necessary amendments were made before extracting data from the included documents. We adopted an inductive approach to extract data from the reviews guided by this review’s aims.

Extracted information included authors’ names and publication years, study locations, review objectives, databases searched, the number of records identified, quality-appraisal methods and tools used, population characteristics of the included studies, types of climate events, mental health outcomes addressed, findings, conclusions, and identified research gaps. Data were extracted by one researcher (LF-TC) and checked by a second researcher (GA-AG).

### 2.4. Data Synthesis

To identify the areas of focus of the research that has been conducted to date, data were organized using framework analysis [18]. The analysis centered on establishing themes that appropriately represented the data. The categories for the framework were determined based on the research questions that guided the review and the information collected from the documents. Categories included descriptions of contexts and/or locations, population characteristics, mental health outcomes addressed, and the type of climate event detailed.

## 3. Results

### 3.1. Article Selection

Our initial search yielded 1286 records. After removing duplicates, we screened 967 titles and abstracts, leading to the exclusion of 926 records that did not meet the inclusion criteria. Subsequently, we conducted a full-text review of 41 articles. Two papers were not retrieved successfully, resulting in 39 articles being assessed for eligibility. Through full-text screening, we excluded 25 papers because they were not reviews (*n* = 18), not related to mental health (*n* = 5), and not related to populations at disproportionate risk (*n* = 2). Ultimately, 14 studies met our inclusion criteria (see Figure 1 for the PRISMA Flow Diagram). The included reviews and their characteristics are listed in Table 2.

### 3.2. Quality Assessment

The methodological quality of the articles was critically appraised in parallel to data synthesis using the Joanna Briggs Institute checklist for systematic reviews [19]. The checklist includes 11 questions that help guide appraisal, and each question was answered as “yes”, “no”, or “unclear”. The assessment was performed by two reviewers (GA-AG and LF-TC). Once the reviewers had reached an agreement, each review was given one point for each positive “yes” answer, a half-point for “unclear” answers, and zero points for “no” answers. This tool was used to assess and report on the reviews’ methodological quality rather than as a screening tool. Overall, the reviews had an average score of 8.6 of 11 (Appendix B). The included reviews’ scores were strengthened by the authors’ clear review questions, clear search strategies, and good data extraction practices, such as using data extraction forms. Reductions in scores were typically related to insufficient quality-appraisal efforts, such as not mentioning quality-appraisal tools or stating how many reviewers performed quality evaluations.

### 3.3. Article Characteristics

The included reviews conducted searches using many different sources including academic research electronic databases such as CINAHL, Cochrane Library, EBSCO, Embase, GeoBase, Google Scholar, Healthstar, Medline, Pandora, and Web of Science. Some authors incorporated additional sources from the grey literature, documentary films, reports from the Red Cross and other international emergency and disaster management organizations, media outlets, and intergovernmental agency reports, including reports from the World Health Organization (WHO) published between 1981 and 2023.

Among the selected reviews, six were scoping reviews [20,21,22,23,24,25], five were systematic reviews [10,26,27,28,29], and three were non-systematic reviews (e.g., narrative and thematic reviews) [30,31,32]. The reviews were published between 2016 and 2023. The number of records included in the reviews varied between 12 records [23] and 294 records [27]. Further information is presented in Table 2.

The studies included in the reviews were conducted in 58 different countries, including multiple entries from the United States (*n* = 7) [10,22,24,25,28,29,32], China (*n* = 6) [20,21,22,24,25,26], Australia (*n* = 5) [10,22,24,25,28], Canada (*n* = 5) [10,22,24,25,29], India (*n* = 5) [20,21,24,26,28], Bangladesh (*n* = 4) [20,22,24,30], the United Kingdom (*n* = 4) [22,24,25,28], Brazil (*n* = 3) [24,25,30], Finland (*n* = 3) [24,27,28], Mexico (*n* = 3) [20,24,26], the Philippines (*n* = 3) [20,21,22], Russia (*n* = 3) [24,27,29], and Sweden (*n* = 3) [24,27,29]. A graphic summary of the locations of the studies included in the reviews is shown in Figure 2.

**Figure 2 ijerph-21-01415-f002:**
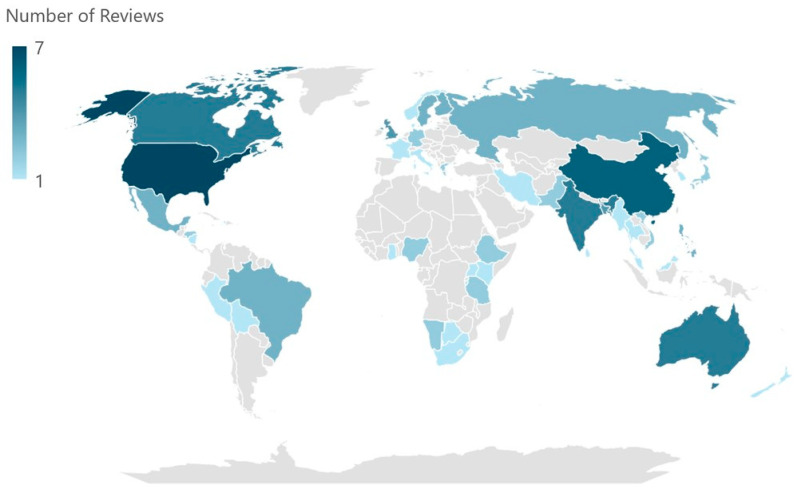
Studies’ locations.

**Table 2 ijerph-21-01415-t002:** Reviews’ characteristics.

Authors	Dates	Source/Database Searched	Research Question/Review Aim	Population Characteristics in Included Studies	Types of Climate Events	Mental Health Outcome Addressed	Number of Records
Batterham et al., 2022 [28]	up until March 2021	PsycINFO, MEDLINE, and Web of Science	What is the effect of environmental factors associated with climate change on rural communities?	Rural residents, people living in LMICs	Access to green space, drought, extreme heat, floods, land degradation	Anxiety, depression, non-suicidal self-injury, substance misuse, suicidal behavior, well-being	28
Berberian et al., 2022 [32]	2017 to 2022	PubMed	What are the effects of extreme weather events on health outcomes for racial and ethnic minority groups, including children, in the USA?	Racial and ethnic groups, children	Extreme cold, extreme heat, floods, hurricanes	Anxiety, depression, PTSD, substance misuse, well-being	89
Deglon et al., 2023 [23]	2008 to 2021	MEDLINE, Academic Search Premier, Africa-Wide Information, CINAHL, Health Source: Nursing/Academic Edition, APA PsycARTICLES and APA PsycINFO, Scopus, Web of Science Core Collection, and SciELO	What are the effects of extreme weather events on the mental health of populations in Africa?	People living in LMICs, including children, women, people of low socioeconomic status, farmers	Brushfires, drought, extreme heat, floods	Anxiety, chronic stress, depression, PTSD/trauma disorders, suicidal behavior	12
Herrán and Biehler, 2020 [30]	2010 to 2020	Google Scholar and grey literature	What are the effects of climate migration on mental health?	Climate migrants	Climate change, droughts, extreme heat	Anxiety, depression, mood disorders, PTSD, suicidal behavior	15
Jaakkola et al., 2018 [27]	1990 to 2017	Ebsco, Academic Search Premiere, PubMed, Web of Science, and Scopus	What are the primary, secondary, and tertiary effects of climate change on the Saami population?	Indigenous people— Saami	Climate change	Acute stress, anxiety, depression, well-being	294
Kelman et al., 2021 [31]	up until June 2020	Google Scholar, MEDLINE, PsycNet, PubMed, Scopus, and Web of Science	What are the effects of climate change in small-island developing states (SIDS) on mental health and well-being?	People living in LMICs	Hurricanes, rising sea levels	Acute stress, depression, PTSD, well-being	45
Lebel et al., 2022 [29]	2010 to 2020	PubMed, PsycInfo, Web of Science, Embase, Cochrane, GeoBase, and CINAHL	What are the effects of climate change on the mental well-being and daily lives of indigenous people in the Circumpolar North?	Indigenous people, specifically those in the Circumpolar North	Access to land (changes in ice), changes in wildlife, unpredictable weather	Anxiety, substance misuse, suicidal behavior, well-being	26
Lindsay et al., 2023 [22]	2002 to 2022	Web of Science, Scopus, PsycINFO, Healthstar, MEDLINE, Embase, and ProQuest	What are the effects of climate events on the health of persons living with disability?	Persons living with disabilities and chronic conditions	Drought, dust storms, extreme cold, extreme heat, floods, hurricanes/typhoons, wildfires	Depression, well-being	45
Rataj et al., 2016 [26]	Up to 2014	MEDLINE, Embase, Web of Science, PsycINFO, CAB Direct, PILOTS, Google Scholar, the WHO’s Virtual Health Library (VHL), and Manual searches in *Global Environmental Change* and *Climatic Change*	What are the effects of extreme weather events on the mental and physical health of people living in LMICs?	People living in LMICs	Cyclones/hurricanes, floods, snowstorms, tornados	Anxiety, depression, PTSD	17
Sahu et al., 2023 [24]	2002 to 2022	MEDLINE/PubMed, Google Scholar, and Cochrane Library	What are the effects of climate change on the mental health of indigenous people globally?	Indigenous people	Access to land (changes in ice), unpredictable weather	Anxiety, well-being	29
Sharpe and Davison, 2021 [20]	2007 to 2019	MEDLINE, Embase, APA PsycInfo, Global Health, Sociological Abstracts, grey literature, and Manual searches of the 2019 volumes of the following journals: *International Journal of Environmental Research and Public Health*, *Global Environmental Change*, *Climatic Change*, *Disaster Medicine and Public Health Preparedness*, and *Environmental Health*	What are the primary and secondary effects of climate change and climate events on the mental health of people living in LMICs?	People living in LMICs	Drought, extreme heat, floods, hurricanes/typhoons snowstorms	Acute stress, anxiety, depression, PTSD, substance misuse	58
Sharpe and Davison, 2022 [21]	2007 to 2019	MEDLINE, Embase, APA PsycInfo, Global Health, Sociological Abstracts, Grey literature, and Manual searches of the 2019 volumes of the following journals: *International Journal of Environmental Research and Public Health*, *Global Environmental Change*, *Climatic Change*, *Disaster Medicine and Public Health Preparedness*, and *Environmental Health*	What are the effects of climate events on the mental health of children living in LMICs?	Children living in LMICs	Cyclones/typhoons/tropical storms, floods, rainstorms, snowstorms, tornados	Acute stress, anxiety, depression, PTSD	23
Varshney et al., 2023 [10]	2006 to 2022	MEDLINE, Embase, CINAHL, and PsycInfo	What are the effects of drought and fires on the mental health of populations at disproportionate risk?	Rural residents, people of low socioeconomic status/unemployed, sexual/gender minorities, ethnic minorities, people with low educational attainment, people with pre-existing health conditions, indigenous populations	Drought, fires	Anxiety, depression, psychosis, PTSD, substance misuse, suicidal behavior	18
Woodland et al., 2023 [25]	2000 to 2023	PsycInfo, Embase, and MEDLINE	What are the effects of climate events on the mental health of people with pre-existing mental health conditions?	People with pre-existing mental health conditions	Cyclones/hurricanes, drought, extreme heat floods, wildfires	Anxiety depression, PTSD, substance misuse	31

The reviews reported on a wide range of climate events. We categorized the events into environmental impacts, extreme weather events and phenomena, and general climate change, as depicted in Table 3. This categorization was performed based on the analysis of the included reviews, and discussion between two researchers (GA-AG and LF-TC) following the principles of framework analysis [18].

The categorization of the mental health outcomes was performed according to the guidance of the Diagnostic and Statistical Manual of Mental Disorders 5 (DSM-5) Classification. We categorized the mental health outcomes into groups including anxiety disorders, mental health and well-being, mood disorders, schizophrenia spectrum and other psychotic disorders, substance use disorders, suicidal behavior/non-suicidal self-injury, and trauma- and stressor-related disorders. The studies are shown in Table 4.

### 3.4. Main Populations at Disproportionate Risk of Health Impacts and Inequities

The reviews focused on several different populations at disproportionate risk of health impacts and inequities. Five reviews examined the mental health of people living in low-and-middle-income countries (LMICs) [20,23,26,28,31], with one review specifically focused on children in LMIC contexts [21]. While not all people living in LMICs are at a disproportionate risk of health impacts and inequities, the populations living in LMICs highlighted in these reviews were disproportionately affected by climate-change-related events [20,23,26,28,31], and there was a reported lack of mental health service capacity to support the community’s needs, particularly in response to acute climate events. Furthermore, many children living in LMICs face disproportionate risks related to climate-change-related events, where situations such as environmental disasters can put them at risk of exploitation or abuse [23].

Four reviews examined effects on the mental health of indigenous people [10,24,27,29], who have been affected in numerous ways by climate change. For example, for the Saami communities living in the Arctic (northern parts of Norway, Sweden, Finland, and the Kola Peninsula in Russia), climate change and the resulting changing weather patterns have disrupted cultural practices in this community (such as reindeer herding and ice fishing), which is associated with anxiety and depression [27]. Lebel et al. [29] reported similar results in the Inuit, Inupiat, and Gwich’in communities in the Circumpolar North. In these communities, activities closely connected to the ice and land are interrupted by changes in the ground due to changing temperatures, disrupting access and causing accidents. The forced avoidance of culturally important land has also reduced traditional knowledge and eroded cultural skills [29] and may result in a decreased sense of belonging to their communities [7].

Two reviews examined people living in a rural or remote areas [10,28], who were reported to be disproportionately affected by elevated mental health conditions linked to severe climate events, potentially due to a lack of resources and infrastructure to support mental health needs in rural communities. These results were consistent across a number of countries, including Australia, Canada, Finland, Germany, Greece, India, Iran, Japan, the UK, and the USA [10,28].

One review focused on racial and ethnic groups [32], one review focused on people with pre-existing mental health conditions [25], one review focused on persons living with chronic health conditions or persons with disability [22], and one review focused on climate migrants [30]. These studies indicate that all of these groups may be disproportionately at risk of experiencing the mental health effects of climate change, although research into these populations is extremely limited to date.

### 3.5. Gaps Identified in the Literature

Many of the reviews identified important gaps in the literature, and several themes emerged across the reviews. First, a number of methodological gaps were identified. Several reviews mentioned issues regarding the lack of comparisons between exposed and unexposed groups [10,20,21] or the absence of longitudinal studies that compare effects before and after events [20,21,28]. Without these group or before/after comparisons, it is extremely difficult to quantify the effects of climate-related events on mental health. Second, while many studies have focused on the direct effects of climate change on the mental health of populations at disproportionate risk, there have been relatively few studies investigating the longer-term effects of climate change not associated with acute extreme weather, like long-term droughts or reduced biodiversity, or the indirect effects of climate change, such as eco-anxiety and migration [20,21,26,28]. While these effects are harder to study as they do not have a clear start and finish point like many extreme weather events, they may play a large role in mental health and well-being and require further research. Finally, several reviews called for the assessment tools that are used to measure mental health outcomes to be consistent and standardized to allow for comparisons across studies [10,25,26]. These tools also need to be appropriately adapted for use in non-Western cultures [23,26].

Numerous reviews called for more extensive research on the effects of climate change on the mental health of specific groups. While research on indigenous groups was highlighted in several of the reviews [24,27,29], many of these populations face unique effects of climate change due to cultural relationships with the land. As a result, many reviews emphasized the need for further and more in-depth research to understand the specific risk factors for indigenous people and ethnic minorities [23,24,27,28,29,32], as well as people in geographical regions underrepresented in the literature, such as LMICs [20,21,23,26,28,31]. A number of reviews also highlighted a gap in the literature on the effects of climate change on the mental health of people with other protected characteristics, including gender [25,31] and age [22,25,28,31], as well as the intersectionality of characteristics [10,22,31,32]. Finally, climate change is creating climate migrants in many geographical areas, and further research is necessary to understand the effects of climate migration on mental health [30]. The identified gaps and research recommendations are detailed in Table 5.

## 4. Discussion

This rapid scoping review of reviews summarized the research areas to date on how climate change affects the mental health of populations at disproportionate risk of health impacts and inequities and the identified gaps in the literature, which, if addressed, could be beneficial for developing future recommendations and interventions. To the best of our knowledge, this is the first review of reviews exploring this topic in this emerging field.

### 4.1. Critical Gaps in the Field

While this review highlighted the specific impacts of climate change on several populations, including people living in LMICs, indigenous communities, and people living in rural areas, there was also a clear need for further research to address effects on other populations that may be at disproportionate risk. There is a critical lack of studies focusing on other protected characteristics, such as gender, disability, and pre-existing mental health conditions. The intersectionality of these characteristics is also underexplored, which limits the understanding of how multiple risks interact to impact mental health in the context of climate change [10,22,31,32].

Age is also a key factor to consider. Older people may be at an increased risk of mental health issues due to isolation and physical health risks [22]. Children are also particularly susceptible to the mental health impacts of climate change [9,33]. Climate impacts occurring during childhood, such as extreme weather events and displacement, can have long-term effects on mental health, affecting cognitive function [34], and these traumatic events can lead to severe distress, causing children to more frequently experience PTSD and depression [9,33,34]. Moreover, educational system disruptions, school damage, and extreme heat affect children’s learning and overall well-being [9]. The loss of social support networks during disasters may increase their risk of developing mental health issues and impact their ability to cope.

Climate anxiety in young people is not a mental health disorder per se [35]. It can be seen as a normal adaptive response to the threat of climate change and can motivate action against it; however, it may also be a risk factor for generalized anxiety disorder, with a potential bidirectional relationship between climate anxiety and generalized anxiety disorder [36]. This area requires further research to understand these possible cause-effect relationships [35].

While there is a growing body of research on the direct impacts of climate change on mental health, there is a need for further studies to explore and address the long-term consequences of slow and gradual environmental changes resulting from climate change, particularly as these may have consequences for the mental health of populations at disproportionate risk of health impacts and inequities by increasing subclinical mental health issues and reducing well-being [20,21,28]. Previous research has shown that subclinical mental health conditions, when co-existing with other risk factors, can increase the risk of developing clinically diagnosed mental health conditions [37]. Thus, the gradual effects of climate change may be an additional factor that interacts with other risk factors to increase mental health risk. Understanding the broad range of the direct and indirect effects of climate change on mental health is critical to develop effective interventions to support the mental health and well-being of populations at disproportionate risk.

### 4.2. Strategies for Mitigating Mental Health Impacts

In the context of climate change, the increasing occurrence of extreme weather events and environmental disruptions is creating a pervasive sense of uncertainty affecting individuals worldwide, which may contribute to mental health concerns indirectly and over the long-term. Thus, it is crucial to explore strategies that address these mental health challenges. To date, most of the selected reviews primarily emphasize the necessity of more robust studies that investigate the effects of climate change on populations’ mental health [20,21,25,27,28,31,32] rather than proposing specific interventions or policies, indicating that additional research may be necessary before informed interventions can be developed.

A recent report [38] put forward policy-level suggestions, emphasizing the importance of increasing community involvement in policy responses to climate change. This 2023 report on sexual and reproductive health and rights in climate commitments described important participatory processes that engaged individuals, especially from younger demographics, offering a fresh perspective on this issue and reiterating the need for political action [38]. The cited report identified 42 countries that have included young people in their policy responses to climate change, particularly in relation to education and awareness. For instance, in Seychelles, young people were invited to participate in monitoring, reporting, and verifying policy implementation, while in South Sudan, strategies such as organizing climate change discussions in clubs and schools and revising school curricula to include climate change and environmental education were deployed [38]. Empowering young people to participate in climate action may positively impact their mental health [21]. More recently, the Connecting Climate Minds initiative [39] worked in co-production with people with lived experience to develop agendas for different global regions and populations that are at disproportionate risk due to the effects of climate change, such as small farmers and fisher-people. These two reports provide examples of how populations affected by climate change can be supported to develop approaches to support their mental health, which can then be incorporated into policy.

### 4.3. Strengths and Limitations

The choice to use a review-of-reviews method was motivated by the existence of previous reviews in the field and the absence of published work that summarizes the current areas of research and highlights the key areas for future research. We also used two independent reviewers during the search for and screening and quality appraisal of the articles, adhering to rigorous systematic research procedures.

While the use of a rapid design in this review allowed for timely results, it meant that only a limited number of databases were searched in a specific time frame. We also used a limited number of search terms, potentially overlooking some subject headings, keywords, and synonyms. This may have limited our results, particularly in identifying populations at disproportionate risk of health impacts and inequities. We specifically used broad terms related to these populations to determine which populations have been identified in the literature as facing disproportionate risk and to avoid bias toward particular groups. However, this approach may have unintentionally excluded certain populations, particularly in cases where reviews chose not to use the broad terms we identified.

We also focused the search on reviews written in English. While it is not likely that there are many reviews in other languages in this emerging field, research studies and grey literature reports in other languages could have key information from non-Western countries, including LMICs. Finally, while there was no intent to perform a meta-analysis in this review, we also did not comment on the magnitude, robustness, and functional relevance of the mental health outcomes in the reviews analyzed; we only addressed the areas of research that have been focused on to date. As noted in the identified gaps, better measurement and quantification of the mental health outcomes are necessary for this field. As these studies become available, a future review or meta-analysis synthesizing these effects would be valuable to clarify the type and degree of impact of climate change on mental health.

## 5. Conclusions

This rapid scoping review of reviews has highlighted the research areas to date on the impacts of climate change on the mental health of populations at disproportionate risk of health impacts and inequities. While this emerging field continues to grow, there are clearly identified gaps in the research, both in terms of methodology and areas of interest, that are important to address in the coming years to more concretely establish the severity and frequency of these impacts. This research will be critical as the basis for developing solutions to support the mental health of populations at disproportionate risk of health impacts and inequities in response to climate events, and it should be co-produced with these populations and the relevant stakeholders in society.

## Figures and Tables

**Figure 1 ijerph-21-01415-f001:**
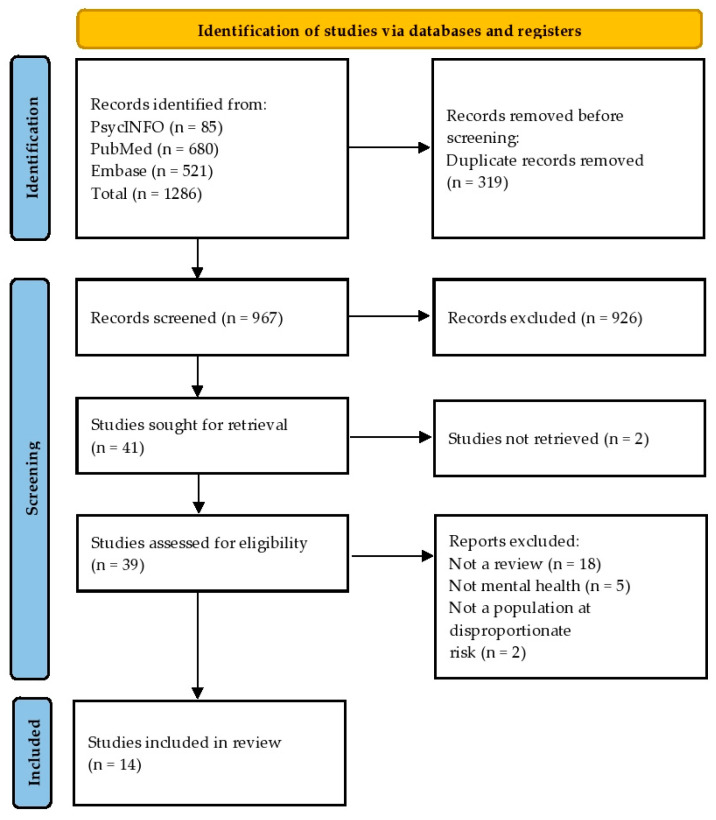
PRISMA flow diagram [12].

**Table 1 ijerph-21-01415-t001:** Inclusion and exclusion criteria.

Inclusion	Exclusion
Literature or evidence review, systematic or non-systematic	Not a review
Available in English	Not pertaining to human health
Describing or mentioning climate-change-related events impacting the mental health of populations at disproportionate risk	Not specifically addressing mental health
Published from 2013 onwards	Focused on the general population’s response to climate change
	Centered on the effects of climate change on animals’ health or the environment

**Table 3 ijerph-21-01415-t003:** Categories and types of climate events.

Category	Type of Climate Event	Authors
Environmental impacts	Access to green space	[28]
	Access to land (including changes in ice)/land degradation	[24,28,29]
	Changes in wildlife	[29]
	Rising sea levels	[31]
	Unpredictable weather	[24,29]
Extreme weather events and phenomena	Cyclones/hurricanes/typhoons/tropical storms	[20,21,22,25,26,31,32]
	Drought	[10,20,22,23,25,28,30]
	Dust storms	[22]
	Extreme cold (including snowstorms)	[20,21,22,26,32]
	Extreme heat	[20,22,23,25,28,30,32]
	Fires (brushfires and wildfires)	[10,22,23,25]
	Floods	[20,21,22,23,25,26,28,32]
	Rainstorms	[21]
	Tornados	[21,26]
General	Climate change	[27,30]

**Table 4 ijerph-21-01415-t004:** Mental health outcomes.

Category	Mental Health Outcome	Authors
Anxiety disorders	Anxiety	[10,20,21,23,24,25,26,27,28,29,30,32]
Mental health and well-being	Mental health and well-being in general	[22,24,27,28,29,31,32]
Mood disorders	Depression	[10,20,21,22,23,25,26,27,28,30,31,32]
	Mood disorders general	[30]
Schizophrenia spectrum and other psychotic disorders	Psychosis	[10]
Substance use disorders	Substance misuse	[10,20,25,28,29,32]
Suicidal behavior/Non-suicidal self-injury	Non-suicidal self-injury	[28]
	Suicidal behavior	[10,23,28,29,30]
Trauma- and stressor-related disorders	Acute stress disorder (ASD)	[20,21,27,31]
	Chronic stress disorders	[23]
	Post-traumatic stress disorder (PTSD)/trauma disorders	[10,20,21,23,25,26,30,31,32]

**Table 5 ijerph-21-01415-t005:** The main research gaps and recommendations discussed in the reviews.

Gap Themes	Recommended Research Approach/Area	Authors
Methodological gaps	Comparisons before and after weather events with longitudinal studies	[20,21,28]
	Comparisons of groups exposed and unexposed to climate events	[10,20,21]
	Consistent use of mental health scales to allow for cross-study comparisons	[10,25,26]
	Development and adaptation of mental health scales appropriate for non-Western cultures	[23,26]
	Measures of long-term and indirect effects of climate change	[20,21,26,28]
Research into additional groups that are underrepresented in the literature	Age	[22,25,28,31]
	Gender	[25,31]
	Indigenous populations and ethnic minorities	[23,24,27,28,29,32]
	Intersectionality between multiple protected characteristics	[10,22,31,32]
	Migrants due to climate change	[30]
	Populations from geographic areas that are underrepresented in the literature, including LMICs	[20,21,23,26,28,31]

## Data Availability

Data are available upon reasonable request from the corresponding author.

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
