# Peer review of "The Impact of Climate Change on the Mental Health of Populations at Disproportionate Risk of Health Impacts and Inequities: A Rapid Scoping Review of Reviews"

_ijerph, 2024, doi:10.3390/ijerph21111415_

Round 1

Reviewer 1 Report

Comments and Suggestions for Authors

This paper presents a rapid review of reviews on the impact of climate change on vulnerable populations. The manuscript is well-written. I have some concerns about the use of the term "rapid review of reviews", in two respects:

- how this is different from a systematic review of reviews and how it differs from a scoping review of reviews. It would be useful to clarify briefly why a rapid review of reviews was undertaken. I can see that some justification is given in "strengths and limitations", but really there should be something in the introduction

- there is lack of clarity and consistency in relation to the review's remit. The aim as stated at the end of the intro is to synthesize the evidence/research gaps. I do not think that the results section really comprises a synthesis as such - Table 4 summarises "mental health outcomes", but there is no indication of how these were assessed - which questionnaire(s)/surveys were used, what were the specific outcomes on each measure/subscale, what were the effect sizes (useful to see even if a meta-analysis isn't conducted).

I would be more enthusiastic about the paper if it was either more comprehensive in providing these details, which would allow for a synthesis of the findings, OR if it rebranded itself, e.g., as a (rapid) scoping review, where the aim is to summarise which outcomes are being looked at in vulnerable groups. At present, I find it a little misleading to say that this is a synthesis of outcomes, without the nuance provided about what the outcomes are and, indeed, where there might not have been significant findings.

The other thing I realised is that the concept of 'vulnerable' is incredibly broad, and a full systematic review might have operationalised vulnerability very carefully and assessed a much broader range of papers (i.e., not even including a 'vulnerable' search term, as papers might not use words like this even though they assess vulnerable groups). I appreciate that this is a rapid review, but perhaps it would be better to include in the introduction whether the need for speed with this review influenced a narrower search strategy. Linked to this, I am not sure it makes sense to discuss non-Western countries in the introduction in relation to vulnerability, and then to conduct a study that does not go out of its way to scope all the literature from non-Western countries - perhaps it is worth removing this from the introduction, or being really clear about whether you consider all non-Western countries to be 'vulnerable' in relation to climate change?

I find the search strategy to be very limited - I would expect terms relating to socio-economic status, low and middle-income countries, ethnicity, i.e., however vulnerability is defined. The paper would need to be much clearer about the reasons for assessing vulnerability by using its specific range of terms, and what the limitations of doing so are - I do not think that any claim can be made that the paper synthesises the research on vulnerable groups given how non-systematic the search stategy is in this regard.

In relation to formatting, in Table 5 it is not clear which authors correspond to which gaps/recommendations. 

Reviewer 2 Report

Comments and Suggestions for Authors

This review of reviews manuscript seeks to synthesize research related to the mental health impacts of vulnerable populations from a global perspective. This is a timely and very relevant topic. The paper is well-described and well-written. A few minor suggestions. 1. There is some discussion about using the term “vulnerable” when describing populations or groups that are at disproportionate risk for health impacts/inequities. You may want to reconsider that terminology. 2. Table 5: currently organized by alphabetical order. It would be more clear to group by recommendation type, for example recommendations for research with Indigenous populations

Round 2

Reviewer 1 Report

Comments and Suggestions for Authors

Thank you - I am satisfied with the authors' responses to my comments. The only final recommendation I have is that the wording of "at disproportionate risk for health impacts..." which is in the title and in various places throughout the manuscript, be changed to "at disproportionate risk of health impacts..." - the second phrasing is present in some places, but it would be helpful for this phrasing to be used consistently throughout.